# *Clostridioides difficile* Infection Rates after Ceftolozane–Tazobactam and Ceftazidime–Avibactam Treatment Compared to Carbapenem Treatment: A Retrospective Single-Center Study

Nagisa Godefroy [1], Helga Junot [2] , Laurence Drieux-Rouzet [3,4] , Cyril Méloni [2], Charles-Edouard Luyt [5,6], Jérôme Robert [3,4] and Alexandre Bleibtreu [1,*]

[1] AP-HP, Hopitaux Universitaires Pitie-Salpetriere Charles Foix, Service de Maladies Infectieuses et Tropicales, 75013 Paris, France; nagisa.godefroy@aphp.fr

[2] AP-HP, Hopitaux Universitaires Pitie-Salpetriere Charles Foix, Pharmacy, 75013 Paris, France; helga.junot@aphp.fr (H.J.); meloni.cyril@gmail.com (C.M.)

[3] AP-HP, Hopitaux Universitaires Pitie-Salpetriere Charles Foix, Bacteriologie—Hygiene Hospitaliere, 75013 Paris, France; laurence.drieux-rouzet@aphp.fr (L.D.-R.); jerome.robert@aphp.fr (J.R.)

[4] Centre d'immunologie et des Maladies Infectieuses CIMI-Paris, INSERM, U1135, 75013 Paris, France

[5] Service de Medecine Intensive Reanimation, Institut de Cardiologie, Groupe Hospitalier Pitie-Salpetriere, Assistance Publique-Hôpitaux de Paris, 47 Boulevard de l'Hôpital, 75651 Paris, France; charles-edouard.luyt@aphp.fr

[6] INSERM, UMRS_1166, ICAN Institute of Cardiometabolism and Nutrition, Sorbonne Universite, 75013 Paris, France

* Correspondence: alexandre.bleibtreu@aphp.fr

**Abstract:** Introduction: Ceftolozane–tazobactam (CT) and ceftazidime–avibactam (CZA) are new beta-lactam/beta-lactamase inhibitors (BL/IBL) and antibiotics. There are few data regarding their impact on *Clostridioides difficile* infections (CDI). The objective of our study was, therefore, to determine and compare the number of CDI occurring after treatment with CT or CZA and carbapenem (CBP). Methods: All patients who received at least one dose of CT or CZA in our hospital between 1 January 2018 and 31 December 2019 were included. We compared, during the same period, the number of CDI after CT or CZA treatment and CBPs by using a chi-square test of Fischer's exact test when required. *p* value < 0.05 was considered as significant. Results: Among the 53 patients receiving CZA and 42 patients receiving CT, two and one, respectively, developed a CDI within 90 days. Of the three (3%) patients who developed a CDI, one died 15 days after his second CDI (36 days after initiation of CZA). Of the 2291 patients receiving CBP, 37 (1.6%) developed a CDI within 90 days. There was no significant difference between the number of CDI occurring after CBP and CT or CZA treatment. CT or CZA use is not associated with an increased rate of CDI compared to CBP.

**Keywords:** Ceftolozane–tazobactam; Ceftazidime–avibactam; *Clostridioides difficile*; gut microbiota; antibiotics

## 1. Introduction

Ceftolozane–tazobactam (CT) and ceftazidime–avibactam (CZA) are newly validated beta-lactam/beta-lactamase inhibitors (BL/BLI). Both provide expansive antimicrobial coverage of Gram-negative bacteria, including *Pseudomonas aeruginosa*, and stable activity against many β-lactamases as well as coverage of most extended-spectrum β-lactamase-producing (ESBL) organisms [1,2]. CZA is also active against carbapenem-resistant *Enterobacteriaceae* that produce Class A carbapenemases such as KPC [3]. However, avibactam does not inactivate metallo-β-lactamases (MBL) such as New Delhi metallo (NDM)-β-lactamases, and, therefore, CZA is of no interest against these strains [4]. Thus, in a context

of increasing prevalence of antibiotic resistance, CT and CZA may represent the last available effective treatments for multidrug-resistant Gram-negative infections. In particular, the place of CT in the treatment of multidrug-resistant *p. aeruginosa* infections is now well recognized [5].

Extensive use of antibiotics has been linked to increasing *Clostridioides difficile* infections (CDI), especially through the modification of the gut microbiota. There are few data concerning the impact of BL/BLI on gut microbiota and CDI, although it has been shown that CDI impacts the length of stay, morbidity and mortality of infected patients [6,7]. CDI incidence has increased in France with 3.6 cases per 10,000 patient-days in French acute healthcare facilities in 2016. According to the French national discharge hospital database, the estimated number of hospital stays with a CDI diagnosis increased from 9270 in 2010 to 19,480 in 2016 (+14%/y). Death occurred in 12% of stays and colectomies were performed in 1% of stays. When restricting to hospital stays with a primary diagnosis of CDI, 6% mortality and 0.4% colectomy were observed [8]. Available data on the relative risk of CDI following antibiotic courses range from 3.2 (1.80–5.71) for third generation cephalosporin, 2.86 (2.04–4.02) for clindamycin and 2.44 (1.32–4.49) for carbapénème [9]. To date, a single study has evaluated the impact of CZA on the gut microbiota in healthy volunteers (HV) [10]. Of the 12 HV receiving CZA for 7 days, five acquired resistant *clostridia* and three resistant *lactobacilli* within 21 days after treatment. To our knowledge there are no published studies on the risk of CDI related to CT or CZA use in real life.

The objective of our study was to determine rates of CDI occurring after treatment with CT or CZA and to compare them with rates of CDI occurring after treatment with carbapenem (CBP).

## 2. Materials and Methods

We performed a retrospective mono center case study over a 2 year period in a Parisian university hospital (1433 beds, 169 ICU beds).

### 2.1. Objective

The main objective of our study was to identify and describe CDI occurring after a CZA or CT use. The secondary objective was to compare number and percent of CDI episodes between CT or CZA and Carbapenem (CBP) groups, and 3 month mortality associated with CDI.

### 2.2. Data Collection

Prescription of Antibiotics

All prescriptions of CZA, CT and CBP are under close supervision in our hospital using a nominative prescription questionnaire and daily delivery with mandatory review for drug continuation at Day 1 by the antimicrobial stewardship team. All patients who received at least one dose of CT or CZA in our hospital between 1 January 2018 and 31 December 2019 were included. Duration and cumulative doses were collected for CT and CZA.

CBP was used as a comparator to CZA and CT because both latter drugs may be used in place of CBP in order to decrease CBP use in the context of increasing CBP resistance, as recommended in France [11].

CDI

A CDI episode was defined as a compatible clinical presentation with the presence of *C. difficile* using the two step algorithm recommended by the ESCMID (antigen + presence of free toxin or positive result for toxin encoding genes by PCR) [12].

CDI was defined as potentially related to CT or CZA when it occurred within 90 days after the first day of treatment [12,13]. Previous history of CDI was also recorded in order to evaluate relapses after treatment. A relapse of CDI was defined as the reappearance of a compatible clinical presentation that had disappeared at the end of the treatment, and

the presence of *C. difficile* antigen and free toxin or toxin gene assessed by a PCR at least 10 days after the last positive sample [14].

Statistical Analysis

A descriptive analysis of the dependent and independent variables was performed. To compare the number of CDI after CT or CZA treatment and CBPs we used a chi-square test or Fischer's exact test when required. $p$ value < 0.05 was considered as significant. All analyses were performed using R Studio Version 1.2.5033.

## 3. Results

Between January 2018 and December 2019, 2386 patients received at least one dose of CT, CZA or CP.

### 3.1. CDI after CZA or CT

During the study period, 53 and 42 patients received CZA and CT, respectively. Among the 53 patients who received CZA, 22 were tested for toxinogenic *C. difficile,* using the ESCMID two step algorithm, because of post antibiotic digestive symptoms and two developed CDI within 90 days (Table 1). One patient (P1) developed a single episode of CDI and the second (P2) two episodes. P1 was diagnosed with CDI 52 days after initiation of CZA. P2 developed a first CDI episode 13 days after initiation of treatment and a first relapse 23 days later, i.e., 36 days after initiation of treatment. P2 died 15 days after the second CDI occurred. The first patient was still alive 3 months after CDI diagnosis. Two patients had a history of CDI before the CZA treatment but did not develop any CDI within the 90 days following treatment initiation.

Eight of the forty-two patients who received CT were tested for toxigenic *C. difficile* for post antibiotic digestive symptoms and only one (P3) developed a CDI within 90 days (Table 1). This occurred 44 days after initiation of treatment. The patient was free of CDI before CT treatment and was still alive 3 months after this CDI episode.

**Table 1.** Characteristics of the 3 patients who developed a *Clostridioides difficile* infection (CDI) after a Ceftolozane–tazobactam or Ceftazidime–avibactam (CZA) treatment.

|  | **Patient 1** | **Patient 2** | **Patient 3** |
|---|---|---|---|
| Treatment received | CZA | CZA | CT |
| Antibiotics dose (cumulative dose) | 54 g | 42 g | 9 g |
| Microbiological documentation | *P. aeruginosa* MDR | *P. aeruginosa* MDR | *P. aeruginosa* MDR |
| Age | 26 | 73 | 75 |
| Sex | M | M | F |
| History of CDI | No | Yes | No |
| PPI * treatment | No | Yes | No |
| Onset of CDI after antibiotics (days) | 52 | 13 and 36 | 44 |
| Status 3 months after CDI | Alive | Dead | Alive |

* PPI: Proton-pump inhibitors.

### 3.2. CDI after CBP

During the study period, 37 of the 2291 patients receiving CBP (1.6%) developed a CDI within the 90 days following treatment initiation. Among the 37 patients, eight (24%) developed multiple CDI episodes. Three had one relapse, two relapsed twice and three had four relapses. Overall, 51 patients had a previous history of CDI before CBP treatment, and six (12%) relapsed in the 90 days after the CBP treatment. CDI occurred a median of 36 days after the first day of treatment. Three months after the CP-related CDI, 15 patients (44%) had died.

### 3.3. Comparison of CBP and CT or CZA

There was no statistically significant difference between the number of CDI occurring after CBP and CT or CZA treatment ($p > 0.05$). (Table 2).

**Table 2.** Characteristics of the patients who developed a Clostridioides difficile infection (CDI) after a carbapenem (CBP), Ceftolozane–tazobactam (CT) or Ceftazidime–avibactam (CZA) treatment.

|  | CZA or CT | CBP |
|---|---|---|
| Number of patients who had the treatment | 95 | 2291 |
| Number of patients who developed a CDI (%) | 3 (3) | 37 (2) |
| Sex ratio M:F | 2:1 | 3.6:1 |
| Mean age [IQR] | 58 [30–86] | 60 [45–75] |
| Mean days between the treatment and the CDI [IQR] | 36 [19–53] | 37 [12–62] |
| Number of patients who had a history of CDI (%) | 2 (2) | 51 (2) |
| Number of patients who relapsed after treatment (%) | 0 | 6 (16) |
| Number of patients who had the treatment | 1 (33) | 15 (40) |

## 4. Discussion

Among the 95 patients who received CZA or CT in our hospital, three patients (3%) developed a CDI within 90 days after antibiotic treatment. This result is far below the five volunteers out of twelve in whom toxigenic *Clostridioides difficile* strains were detected (8). The difference can be explained by the fact that our study looked for episodes of *C. difficile* infection and not for carriage in the stool. Indeed, in the study by Rachid et al., out of the five volunteers with *C. difficile* in the stool, three reported loose stools ( with a duration ranging from one and 10 days), one reported flatulence and one had no clinical signs. These *C. difficile* carriages were therefore not defined as *C. difficile* infections but rather as asymptomatic *C. difficile* colonizations [15]. The prevalence of asymptomatic *C. difficile* colonizations in adults varies according to population groups. In healthy adults up to 55% were colonized by *C. difficile* strains without clinical signs of CDI [16]. If asymptomatic colonization with *C. difficile* (while being toxigenic strains carriers) seems to be a crucial factor in the progression to CDI [17], it does not immediately implicate an infection and its management remains unclear [18]. Based on current information, an eradication of *C. difficile* is not indicated in persons with asymptomatic *C. difficile*. Our study, therefore, provides reassuring results on the safety of using CZA or CT concerning the risk of CDI.

During the study period, 1.6% of patients receiving CBP developed a CDI within the 90 days following treatment initiation; this corresponds with previously published results. Although there are few data on the prevalence of CDI after treatment with CBP, there are studies on the relative risk. Concerning observational studies measuring associations between antibiotic classes and hospital acquired CDI, odd ratios (OR) range from 1.65 (1.01–2.68) [19] to 5.41 (1.38–21.20) [20]. These discrepancies can be explained by many confusing factors involved in the link between antibiotics and CDI. Indeed, sepsis itself has an impact on gut microbiota [21,22], and drugs given in intensive care can also disrupt the gut microbiota [23]. Moreover, within the same class of antibiotics, each molecule does not have the same activity on anaerobes or the same mode of excretion, which makes it difficult to predict the impact on the gut microbiota. Even for the same molecule with biliary excretion, the impact on the gut microbiota varies according to biliary clearance [24]. As shown in our study, 15 patients (44%) who had a CDI after CBP treatment were dead 3 months after the infection, which highlights on the one hand the patients' frailty on which these infections occur and on the other hand the mortality following these infections.

We decided to link an episode of *C. difficile* infection occurring within 90 days after the prescription of CZA or CT to its use because the objective of our study was to look for an increased risk of CDI after CZA and CT prescription. Indeed, we cannot exclude that these patients did not receive other antibiotics during this period. The confounding factors could, therefore, not have underestimated the risk but rather led to an over-risk since these patients had more often received multiple lines of antibiotic therapy [25].

Our work has several limitations. The first limit is the low number of patients treated by CT or CZA over two years in our center, mostly due to the restrictive use of these drugs. French guidelines for broad spectrum antibiotic use place these two BL–IBL associations

as last line therapies and not as carbapenem sparing. The antibiotic stewardship policy in our center strongly monitors CT and CZA use. As stated in Table 2 with three examples of included patients, CT and CZA were only used in MDR bacteria. These bacteria, including *P. aeruginosa*, had to be tested in a bacteriological lab and undergo disk diffusion testing to the BL–IBL association. Without susceptibility determination for the clinical bacteria involved in the infection, the clinician could not use CT or CZA. The second limitation is the absence of systematic testing for *C. difficile* in these patients and the absence of systematic screening for symptoms related to potential CDI due to the non-interventional and retrospective design of our study. Third, the history of previous antibiotic use in included patients was not captured during the medical chart screening. This is an important confounder that could have impacted the occurrence of CDI. Likewise, antibiotic use after CBP or CT or CZA was not captured and we cannot exclude that patients did not receive other antibiotics during the 90 day period. Fourth, clinical conditions and previous antibiotic exposure might impact susceptibility to develop CDI. Underlying conditions requiring extensive antibiotic use may predispose patients to CDI, such as COPD flare-ups. At the opposite end there are therapies that can impact acute exacerbations of an underlying condition that impact the probability for a patient with the same clinical condition to be exposed to more or fewer courses of antibiotics depending on the ancillary therapies he or she undertakes. By extension, the patient will be more or less prone not only to CDI but also to severe forms of CDI because of his or her underlying clinical condition [26]. We did not capture or stratify data to be able to apply an analysis of the propensity of this. Despite these limitations and to progress on this debate, it would be interesting to continue with prospective studies to provide clear answers using systematic visits and feces screening as better evaluations of the use of other drugs which could increase the risks of CDI episodes. Unfortunately, since 5 December 2020, CT availability has been internationally suspended due to *Ralstonia pickettii* contamination in several batches [27]. A global voluntary recall was performed, but there is still no planned date for availability.

## 5. Conclusions

Only three patients (3%) developed a CDI after CZA or CT prescription. There was no significant difference between the number of CDI occurring after CBP and CT or CZA treatment. CDI occurred for CZA, CT and CBP approximately 35 days after the first day of treatment. This study provides reassuring data on CDI risk following CZA or CT use in real life.

**Author Contributions:** Conceptualization, A.B. and N.G.; methodology, A.B. and N.G.; software, H.J., C.M., L.D.-R. and J.R.; validation N.G.; formal analysis, N.G.; investigation, N.G.; writing—original draft preparation, A.B. and N.G.; writing—review and editing, H.J., C.-E.L., C.M., L.D.-R. and J.R.; supervision, A.B.; project administration, A.B. All authors have read and agreed to the published version of the manuscript.

**Funding:** This research received no external funding.

**Institutional Review Board Statement:** Ethical review and approval were waived for this study due to it being retrospective non-interventional and on file character. Since our study concerns the secondary reuse of data collected within the patient's management, without collecting new data, it is in line with the reference methodology MR-004 and is exempt from additional regulatory authorizations in France.

**Informed Consent Statement:** Patient consent was waived due to systematic information in entrance to our institution.

**Data Availability Statement:** Data is available on demand to alexandre.bleibtreu@aphp.fr.

**Conflicts of Interest:** A.B. declares links of interest with Pfizer, Shionogi, Eumedica, Sanofi, Gilead, Astellas, Menarini, Janssen for congress fees, symposia or investigator in clinical studies. The other authors declare no conflict of interest.

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
