# Peer review of "Clostridioides difficile Infection Rates after Ceftolozane–Tazobactam and Ceftazidime–Avibactam Treatment Compared to Carbapenem Treatment: A Retrospective Single-Center Study"

_2673-947X, doi:10.3390/hygiene1030009_

Round 1

Reviewer 1 Report

The authors conduct a retrospective study on the rate of Clostridioides difficile infection after

treatment with Ceftolozane-tazobactam and Ceftazidime-avibactam.

The introduction seems too brief and so much so that it does not allow for an adequate framing of the problem.

In the materials and methods section, I believe that lines 57-70 should be removed.

The topic covered is of current great interest, and the purpose of the study is well described and is clear.

With regard to the statistical analysis, in my opinion, possible confounding factors have not been adequately taken into account. Nor do risk adjustment factors seem to be used to stratify patients on the basis of previous clinical conditions and possible use of other drugs, supplements, which could increase the risk, but, in theory, also decrease it. There are, in fact, clinical conditions requiring extensive antibiotic use that may predispose to CDI, such as COPD flare-ups. It has also been demonstrated that there are therapies that can reduce COPD exacerbations, which is why a patient with the same clinical condition may use more or fewer courses of antibiotics depending on the ancillary therapies he or she undertakes and thus be more or less prone not only to CDI, but also to severe forms of CDI because of his or her underlying clinical condition.

To this end, I recommend reading:

  • Troiano G., Messina G., Nante N. Bacterial lysates (OM-85 BV): a cost-effective proposal in order to contrast antibiotic resistance. J PREV MED HYG 2021; 62: E564-E573 https://doi.org/10.15167/2421-4248/jpmh2021.62.2.1734

Nor does it appear stratified by clinical severity of presentation following administration of the three different antibiotics studied.

With regard to taking other concomitant therapies, the same authors (line 176) state that they cannot rule out the possibility that these patients took other antibiotics during the same period and, at line 177, state that, even if they did, it should not have changed the risk of CDI after the use of CZA or CT: authors should add a literature reference to support this statement.

The limits are, therefore, only partly described.

Finally, in the conclusions, the three objectives described in the materials and methods section are not fully reported.

Author Response

We thank the reviewer for his comments. Please find attached the responses to your comments.

Reviewer 2 Report

Mild improvements in English language throughout the text are required.

Authors need to clear out the parts that are included as a template.

I believe the title needs modification, to include the comparison to CDI rates in patients who received carbapenems and also to avoid conclusions that might be based on study limitations or confounders. A suggestion: "Clostridioides difficile infection rates after Ceftolozane-tazobactam and Ceftazidime-avibactam treatment compared to Carbapenem-treatment: a retrospective single-center study"

Abstract: conclusion missing.

Intro-Methods

Lines 52-55 and 73-74 (study objective): I believe the objective should be in rates than in numbers.

Lines 56-72: this is from the template, please remove.

Please further describe the study setting.

Line 90: is there a definition of CDI in France based on antigen detection? For example, to the best of my knowledge, the IDSA guidelines require other diagnostic methods (eg.toxin detection) additional to antigen detection. This is an important methodologic issue and needs clarification to further support the study results and conclusions. To this end, please also mention in the results, the method by which CDI patients were diagnosed.

Line 92: “within” 90 days

Results

Is a history of previous antibiotics available in included patients? This is an important confounder and could have impacted the occurrence of CDI.

Table 2 last line: please remove the denominator, as it is already mentioned in the first line.

Discussion

Lines 169-172: please cite this statement. I’m not certain whether death 3 months after CDI implies CDI-related mortality; it could reflect other factors, such as frailty (preferrable to the word “fragility” and other comorbidities). Please elaborate.

Line 173: “within” 90 days

Line 181: please define BL-IBL

Line 184: please correct politic to policy.

Another limitation is the limited patient information included, thus limiting the strength of findings and conclusions, due to limited analyses.

Conclusions

Please redefine and make more formal (eg.remove the phrase “as always).

Lines 200-201, 207, 212-216, 217-218: please correct

Can authors explain why ethical approval was waived? Is there a relevant decision?

Author Response

Mild improvements in English language throughout the text are required.

We thank the reviewer for his proposal, the manuscript was reviewed by one of our collaborators who is a native English speaker

Authors need to clear out the parts that are included as a template.

We thank the reviewer for his extensive review,  template parts were removed from the new manuscript

I believe the title needs modification, to include the comparison to CDI rates in patients who received carbapenems and also to avoid conclusions that might be based on study limitations or confounders. A suggestion: "Clostridioides difficile infection rates after Ceftolozane-tazobactam and Ceftazidime-avibactam treatment compared to Carbapenem-treatment: a retrospective single-center study"

We thank the reviewer and we agree with this advice. The new title of the manuscript is now:” Clostridioides difficile infection rates after Ceftolozane-tazobactam and Ceftazidime-avibactam treatment compared to Carbapenem-treatment: a retrospective single-center study”

Abstract: conclusion missing.

We thank the reviewer, a conclusion was added : “CT or CZA use is not associated with increase rate of CDI compare to CBP”.

Intro-Methods

Lines 52-55 and 73-74 (study objective): I believe the objective should be in rates than in numbers.

We thank the reviewer and we replaced numbers by rates

Lines 56-72: this is from the template, please remove.

We thank the reviewer for his extensive review,template parts were removed from the new manuscript

Please further describe the study setting.

Our study was conducted in a Parisian university hospital (1433 beds, 169 ICU beds).

Line 90: is there a definition of CDI in France based on antigen detection? For example, to the best of my knowledge, the IDSA guidelines require other diagnostic methods (eg.toxin detection) additional to antigen detection. This is an important methodologic issue and needs clarification to further support the study results and conclusions. To this end, please also mention in the results, the method by which CDI patients were diagnosed.

We thank the reviewer for this comment. The definition of CDI in France is the same as in ESCMID guidelines. Symptoms of CDI + positive stool sample using a two-step algorithm ( antigen detection + free toxin detection or antigen + PCR screening for toxin gene using geneXpert method for ToxA, Toxin B or TcD.

Information on guidelines and algorithm are now added.

Line 92: “within” 90 days

Within is now added 

Results

Is a history of previous antibiotics available in included patients? This is an important confounder and could have impacted the occurrence of CDI.

We thank the reviewer for this comment. These data were not captured in our work. We agree that exposition to antibiotics including spectra, duration, and route of administration may influence the CDI incidence. We think that it will not change the message of this work which is a loweroccurrence of CDI episodes after CT or CZA than published in healthy volunteers. History of antibiotics before or after CT/CZA or BCP might impact our result. But more toward an CDI overestimation  than an underestimation,except if patients received metronidazole oral course. However, we added this limit in the discussion of the revised manuscript“

Third, history of previous antibiotics use in included patients was not captured during the medical chart screening. This is an important confounder and could have impacted the occurrence of CDI. This the same for antibiotic use after CBP or CT or CZA use because we cannot exclude that these patients did not receive other antibiotics during the 90 days period.

Table 2 last line: please remove the denominator, as it is already mentioned in the first line.

Table was redrawn as suggested by the reviewer

CZA or CT

CBP

Number of patients who had the treatment

Number of patients who developed a CDI (%)

Sex ratio M:F

95

2291

3 (3)

37 (2)

2:1

3.6 :1

Mean age [IQR]

Mean days between the treatment and the CDI [IQR]

58 [30-86]

60 [45-75]

36 [19-53]

37 [12-62]

Number of patients who had a history of CDI (%)

Number of patients who relapsed after treatment (%)

Number of patients who had the treatment

2 (2)

51 (2)

0

6 (16)

1 (33)

15 (40)

Discussion

Lines 169-172: please cite this statement. I’m not certain whether death 3 months after CDI implies CDI-related mortality; it could reflect other factors, such as frailty (preferrable to the word “fragility” and other comorbidities). Please elaborate.

We chose the duration of 3 months to be in line with the largest study analysis of patients with a fatal outcome of CDI : Czepiel et al. Mortality Following Clostridioides difficile Infection in Europe: A Retrospective Multicenter Case-Control Study. Antibiotics (Basel). 2021 Mar.

Line 173: “within” 90 days

Within is now added 

Line 181: please define BL-IBL

We defined it : betalactam / betalactamase inhibitors (BL/BLI).

Line 184: please correct politic to policy.

The change was done as proposed by the reviewer

Another limitation is the limited patient information included, thus limiting the strength of findings and conclusions, due to limited analyses.

We to assess this fact in the limitation part of the discussion. Conclusions

Please redefine and make more formal (eg.remove the phrase “as always).

Final sentence was modified.

Lines 200-201, 207, 212-216, 217-218: please correct

Can authors explain why ethical approval was waived? Is there a relevant decision?

Since our study concerns the secondary reuse of data collected within the patient's management, without collecting new data, it is in line with the reference methodology MR-004 and is exempt from additional regulatory authorizations in France.

Reviewer 3 Report

The author presented a brief report determined the number of Clostridioides difficile infections (CDI) occurring after treatment with Ceftolozane-tazobactam (CT) or ceftazidime-avibactam (CZA) and to compare them with the number of CDI occurring after treatment with carbapenem (CBP). The authors conclude there was no significant difference between the number of CDI occurring after CBP and CT or CZA treatment.

As author stated in the discussion, the sample size is small and more studies needed to validate the finding. Due to small sample size and simple objective, suggest change the manuscript type as Short report or Brief report. 

Other minor points: 

Please delete the information not related to this manuscript from the template. 

Line 24: Move word "respectively" to the end of the sentence.

Line 135: "Table 1" is repeated twice. Please delete one of them.

In Table 1, Microbiological documentation, all three patients developed CDI also co-infected with P. aeruginosa MDR. The impact of this co-infection also needs to be discussed in the discussion section.

Line 151: Authors mentioned "in some studies", however, only cite one study. Not consistent.

Author Response

Reviewer 2

Comments and Suggestions for Authors

The author presented a brief report determined the number of Clostridioides difficile infections (CDI) occurring after treatment with Ceftolozane-tazobactam (CT) or ceftazidime-avibactam (CZA) and to compare them with the number of CDI occurring after treatment with carbapenem (CBP). The authors conclude there was no significant difference between the number of CDI occurring after CBP and CT or CZA treatment.

As author stated in the discussion, the sample size is small and more studies needed to validate the finding. Due to small sample size and simple objective, suggest change the manuscript type as Short report or Brief report. 

Thanks to the reviewer for this point. May the editor gives his advice on this point. First reviewer did not address this point.

Other minor points: 

Please delete the information not related to this manuscript from the template. 

We thank the reviewer for his extensive review and template part were removed from the new manuscript

Line 24: Move word "respectively" to the end of the sentence.

We thank the reviewer, it is now removed from the revised manuscript

Line 135: "Table 1" is repeated twice. Please delete one of them.

In Table 1, Microbiological documentation, all three patients developed CDI also co-infected with P. aeruginosa MDR. The impact of this co-infection also needs to be discussed in the discussion section.

We thank the reviewer for this comment.  The 3 patients with CDI had pulmonary infection due to P. aeruginosa. We did not know if they were also colonized with P. aeruginosa in their gut microbiota. It is known that gut microbiota in CDI patient is less diverse and exhibit an increase in Protoebacteria but to our knowledge impact of P. aeruginosa gut colonization in CDI was not documented. However, we added this part in the discussion of the revised manuscript:

‘As stated in Table 2 with three examples of included patients CT and CZA were uses only in MDR bacteria. These bacteria including P. aeruginosa had to be tested in bacteriological lab and be considered susceptible with disk diffusion testing to the BL-IBL association. Without susceptibility determination for the clinical bacteria involved in infection clinician could not use CT or CZA’.

Line 151: Authors mentioned "in some studies", however, only cite one study. Not consistent

Some studies was removed..

Round 2

Reviewer 1 Report

The authors have made the requested revisions and improved the manuscript, providing high proof of work.

The changes made have improved the soundness of the research, so I consider that the manuscript can be published

Author Response

We thank the reviewer for his comment.

Reviewer 2 Report

Most of my previous remarks are addressed in the revision.

Some minor comments:

Please add the ESCMID reference.

Line 168: “Forty percent of patients who had a CDI after CBP treatment are dead 3 months after the infection, which highlights”. This is a remark I also made during the previous round of review. Please add reference.

Line 208 (in reference to Ralstonia contamination of batches): please add reference.

Conclusions: please rephrase completely without ambiguous statements. What did your study add to the literature and what gaps are left for further studies? “prospective studies to provide clear answers are needed.” is not a clear concluding remark.

Author Response

We thank the reviewer for his comments

  • We had the ESCMID (Crobach et al.) reference
  • The 40% provide from our study. With 15 patients (44%) dead after CDI in the carbapenem group
  • Reference is added
  • Conclusion is modified now